# Object Detection in Drone Video with Temporal Attention Gated Recurrent Unit Based on Transformer

**Zihao Zhou [1], Xianguo Yu [2,\*] and Xiangcheng Chen [3]**

[1]  School of Automation, Wuhan University of Technology, Wuhan 430070, China; stars_zihao@whut.edu.cn
[2]  College of Intelligence Science and Technology, National University of Defense Technology, Changsha 410073, China
[3]  School of Artificial Intelligence, Anhui University, Hefei 230039, China; chenxgcg@ustc.edu
[\*]  Correspondence: yuxianguo11@nudt.edu.cn; Tel.: +86-1887-4883-361

**Abstract:** Unmanned aerial vehicle (UAV) based object detection plays a pivotal role in civil and military fields. Unfortunately, the problem is more challenging than general visual object detection due to the significant appearance deterioration in images captured by drones. Considering that video contains more abundant visual features and motion information, a better idea for UAV based image object detection is to enhance target appearance in reference frame by aggregating the features in neighboring frames. However, simple feature aggregation methods will frequently introduce the interference of background into targets. To solve this problem, we proposed a more effective module, termed Temporal Attention Gated Recurrent Unit (TA-GRU), to extract effective temporal information based on recurrent neural networks and transformers. TA-GRU works as an add-on module to bring existing static object detectors to high performance video object detectors, with negligible extra computational cost. To validate the efficacy of our module, we selected YOLOv7 as baseline and carried out comprehensive experiments on the VisDrone2019-VID dataset. Our TA-GRU empowered YOLOv7 to not only boost the detection accuracy by 5.86% in the mean average precision (mAP) on the challenging VisDrone dataset, but also to reach a running speed of 24 frames per second (fps).

**Keywords:** drone video object detection; deformable transformer; recurrent neural network; feature aggregation

## 1. Introduction

Recently, computer vision researchers are becoming more and more interested in the field of video object detection (VOD). Images obtained by moving platforms often suffer from appearance deterioration due to motion blur, partial occlusion, and rare poses, especially by unmanned aerial vehicles (UAV). These issues have hindered advanced image-based object detectors from reaching a higher standard for challenging real-world scenarios such as drone vision.

Previous VOD methods [1–4] attempted to leverage the rich temporal and motion context in videos. Some methods [4,5] utilize the motion information extracted by an extra optical flow net to guide feature fusion. However, it is difficult to obtain accurate flow features for videos. In contrast, some methods [6,7] attempt to exploit the video context by long short-term memory networks (LSTM). LSTM combines temporal features from different video frames with a forgetting gate and an update gate. Nevertheless, when it comes to UAV images, LSTM-based VOD methods are proven to introduce a significant amount of noise into targets due to the rapid changes in appearance and the small size of objects in drone footage. Other approaches [1,8] leverage deformable convolution to estimate object motion and utilize the displacement to align features in multiple frames. Recently, the transformer model has been utilized to learn video context features for object detection [3], and this method obtains state-of-the-art results on the ImageNet-VID dataset.

Deformable Detection Transformer (Deformable DETR) is employed in this method to boost object detection performance on drone videos.

Our philosophy is to acquire rich, high accuracy while maintaining a lower computation burden. We made some tune-ups according to previous works on video object detection. We chose Gated Recurrent Unit (GRU), Deformable Detection Transformer (Deformable DETR), DeformAlign module, and temporal attention and fusion module to compose our temporal processing module. The proposed method captures temporal context from known video frames to enhance target features in the current frame. To achieve this, we employ the GRU to fuse features of different images. Notably, we depart from the commonly used Convolutional Gated Recurrent Unit (Conv-GRU) and instead use a Deformable Detection Transformer (Deformable DETR) in place of convolution. This modification enables our network to better focus on areas relevant to the targets being detected. Additionally, drone videos often suffer from significant degradation in visual quality; we employ deformable convolution to effectively learn the deviations between the features of reference frame and temporal features. This process enables us to accurately align the temporal features with the reference frame features by taking into account the offsets. Notably, the frames that are most relevant to a given reference frame are probably its immediate neighbors. To reflect this, neighboring frames are always assigned with higher weights than frames far from the reference frame in the proposed temporal attention module. We then use a weighted fusion method to combine the aligned temporal features with the reference frame features, resulting in a set of fused features that are subsequently fed into the detection network to generate detection results for the reference frame.

The main contributions are summarized as follows:

- We proposed an effective and efficient TA-GRU module to model the temporal context between videos. It is very effective to handle appearance deterioration in drone vision, and it can easily be used to promote existing static image object detectors to effective video object detection methods.
- We proposed a new state-of-the-art video object detection method, which not only achieves top performance on the VisDrone2019-VID dataset, but also runs in real-time.
- Compared to previous works, we integrated recurrent neural networks, transformer layers, and feature alignment and fusion modules to create a more effective module for handling temporal features in drone videos.

## 2. Related Work

Image-based Object Detection: Image-based detectors can be categorized broadly into two groups: two-stage detectors and one-stage detectors. Two-stage detectors first generate region proposals and then refine and classify them. Some representative methods in this category include R-CNN [9], SSD [10], RetinaNet [11], Fast RCNN [12], and Faster R-CNN [13]. While two-stage detectors tend to be more accurate, they are also slower. On the other hand, one-stage detectors are usually faster but less accurate, as they directly predict the region proposals based on the feature map. Relevant research in the field of object detection includes various iterations of the YOLO series, such as YOLOv5 [5], YOLOX [14], and YOLOv7 [6]. In our work, we utilized YOLOv7 as the base detector and extended its capabilities for video object detection.

Video Object Detection: Compared to image object detection, video object detection provides more comprehensive information about targets, including motion and richer appearance details. In recent years, researchers have tried to utilize neighboring frame features to enhance reference frame features. However, the presence of varying offsets in each frame poses a significant challenge to effectively utilizing these features. Previous studies attempted to address this issue by aligning the neighboring frame features with the reference frame features. Alternatively, some methods choose to overlook the offsets in each frame and instead use specialized modules to extract temporal information from videos.

Feature Aggregation: To address the issue of significant degradation in the visual quality of drone videos, various previous methods focus on feature aggregation. This

technique involves enhancing the reference features by combining the features of adjacent frames. For instance, FGFA [4] and THP [15] utilize the optical flow produced by FlowNet [14] to model motion relations and align various frames. Alternatively, the optical-flow-based framework [5] categorizes video images based on the background, acquires the optical flow of the input sequence using FlowNet [14], and eventually aggregates the optical flow to model motion relations. Nevertheless, flow-warping-based techniques have some drawbacks. Firstly, drone videos frequently comprise numerous small objects, which make it challenging to accurately extract optical flow. Secondly, it is important to note that obtaining optical flow demands a considerable amount of computational resources, which can make real-time detection a challenging task. In contrast, some other approaches employ deformable convolution to compute the offsets in different frames. This method allows for the adaptive adjustment of convolutional kernel parameters to obtain corresponding offsets. For instance, STSN [8] utilizes stacked 6-layer deformable convolutional layers to gradually aggregate the temporal contexts. TCE-Net [1] takes into account that the contribution of neighboring frames to the reference frame may differ. To align frames, it uses a single deformable convolutional layer and a temporal attention module, which assigns weights to frames based on their respective contributions. However, the task of drone video object detection presents significant challenges, and relying solely on a single deformable convolutional layer can make it difficult to accurately compute the offset between neighboring frames and the reference frame. To avoid introducing excessive computation, simply increasing the number of deformable convolutional layers is not the ideal solution. Our approach, however, is to utilize the GRU module in our TA-GRU to transfer temporal features and incorporate a temporal context enhanced aggregation module to obtain the fusion features that are then fed to the detection network. This method allows us to avoid the need for aligning every neighboring frame with a reference frame and instead adopt a frame-by-frame alignment strategy, which not only reduces computation but also enhances alignment accuracy.

Some recent studies have utilized recurrent neural networks, such as long short-term memory networks (LSTM), to propagate temporal features that contain previous video features. STMN [6] and Association LSTM [7] attempt to model object association between different frames by applying LSTM or its variants. However, the object association modeled by these methods is often imprecise, particularly in drone videos. On the other hand, Conv-GRU utilizes convolution to replace linear calculation, which introduces significant challenges to the GRU module originally used for calculating sequences. TPN [16] adopts a unique method of object tracking which differs from general video object detections. The proposed approach involves linking multiple frames of the same object to generate a segment of tube, which is then fed into an ED-LSTM network to capture temporal context. However, this method introduces significant background noise that can compromise the accuracy of the results. To address this issue, recent research has explored the use of transformers for video object detection. TransVOD [3] demonstrated that incorporating self-attention and cross-attention modules can improve the model's focus on the target regions. Building on this work, our TA-GRU method aggregates temporal features and applies deformable attention instead of convolution to enhance performance. We elaborate on the details of TA-GRU in Section 3.

## 3. Proposed Method

To enable both high accuracy and high efficiency for UAV based image object detection, we proposed a new, highly effective video object detection framework termed TA-GRU YOLOv7. Particularly, we designed four effective modules including, the Temporal Attention Gated Recurrent Unit (TA-GRU) to enhance attention to target features in the current frame and improve the accuracy of motion information extraction between frames; the Temporal Deformable Transformer Layer (TDTL) to reduce additional computational overhead and strengthen the target features; a new deformable alignment module (DeformAlign) to extract motion information and align features from two frames; as well as a temporal

attention based fusion module (TA-Fusion) to integrate useful information from temporal features into the current frame feature.

### 3.1. Overview of TA-GRU YOLOv7

YOLOv7 is one of the most popular image object detectors, at present, due to its great balance between speed and accuracy. Compared with the previous YOLO structures, the backbone of YOLOv7 has a more intensive hop connection structure, which makes it possible to extract richer and more diverse features from the input image. At the same time, it uses an innovative downsampling structure that could reduce the number of parameters while maintaining high accuracy, making it highly efficient and effective. It uses max pooling and features with a step size of $2 \times 2$ for parallel extraction and compression. As a mature and representative image object detector, YOLOv7 has already reached a performance bottleneck. Therefore, we selected it as the baseline for our study on how to effectively enhance the performance of current single frame image detection algorithms in the context of video object detection problems.

The architecture of TA-GRU YOLOv7 is illustrated in Figure 1, which takes multiple frames of a video clip as inputs and generates detection results for each frame as outputs. In order to make sure that our TA-GRU module can be handily applied to various single frame image detectors, we retained all structures of YOLOv7 and only added our TA-GRU module at the neck of YOLOv7, which serves to confirm the efficacy of our module. Our TA-GRU module contains four main components: Temporal Attention Gated Recurrent Unit (TA-GRU) to propagate temporal features, Temporal Deformable Transformer Layer (TDTL) to increase the attention on target regions, DeformAlign to model object motion and align the features from frame-to-frame, and temporal attention and temporal fusion module (TA-Fusion) to aggregate features from videos.

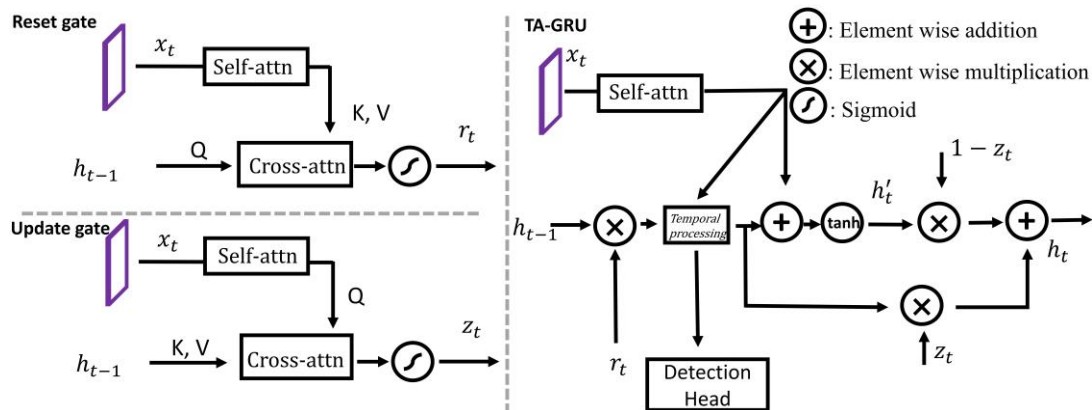

**Figure 1.** Architecture of TA-GRU. In TA-GRU, input features $x_t$ interact with temporal features $h_{t-1}$ through a temporal processing module (temporal alignment and fusion) to obtain enhanced features to feed to the detection head. Additionally, temporal features $h_{t-1}$ will be updated by update gate features $z_t$, where *self_attn* is self-deformable transformer layer, *cross_attn* is cross-deformable transformer layer.

Analysis of Model Complexity. Here, we analyzed the model complexity of our proposed modules to the existing object detectors. These methods have two main computational loads: 1. feature extraction network from the backbone $C_{backbone}$; 2. detection head $C_{head}$. Therefore, the total computational complexity is $O(C_{backbone} + C_{head})$.

In our proposed models, we introduced a simple but effective module $C_{temporal}$ to extract the temporal information in drone videos. Therefore, during the training process, the computational complexity of our model is defined as $O\left(C_{backbone} + C_{head} + C_{temporal}\right)$. We only increased the computational overhead required for the temporal processing module. Adding our module only increases the parameters of the model from 37,245,102 parameters

to 46,451,727 parameters. However, it surpassed the baseline in terms of detection accuracy by a significant margin.

### 3.2. Model Design

Convolutional Gated Recurrent Unit (Conv-GRU). Gated Recurrent Unit (GRU) neural network is a recurrent neural network, a variant of LSTM. Based on LSTM neural network, the cell structure is optimized to reduce parameters and accelerate training speed. The overall Convolutional Gated Recurrent Unit (Convolutional GRU) architecture is shown in Figure 2. There, $x_t$ is the feature extracted by backbone, which uses convolution to compute the update gate and reset gate to renew the temporal feature $h_{t-1}$. However, this temporal feature contains a lot of background from previous frames due to the complexity of drone images. To solve this problem, we aimed to incorporate additional modules into the Conv-GRU architecture to perform deeper temporal feature processing and improve its attention towards the target region. Additionally, inspired by TransVOD [3], we explored the effectiveness of deformable transformer layers in drone video object detection tasks. Based on our experiments, incorporating deformable transformer layers prior to subsequent temporal feature processing enables our network to effectively concentrate on the target area and, to a certain extent, mitigates the impact of background information on the temporal feature processing stage.

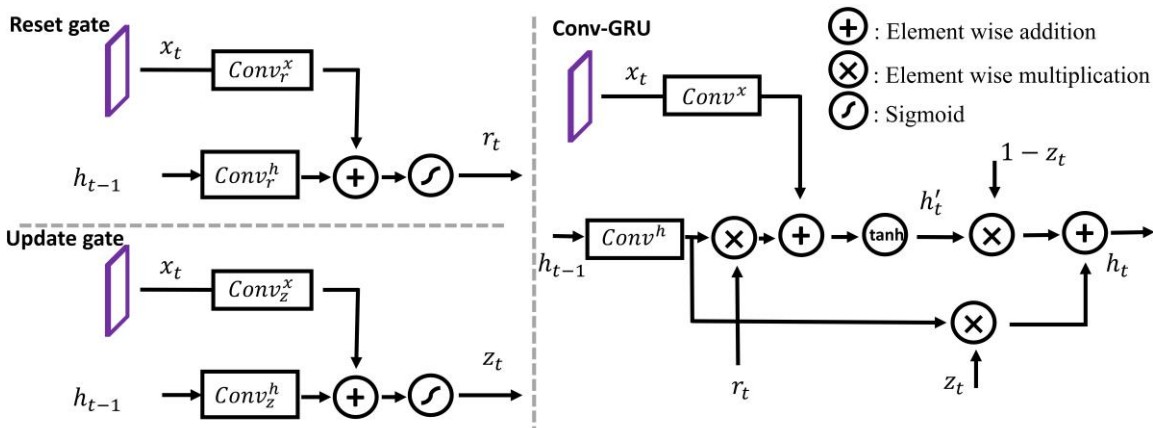

**Figure 2.** Architecture of Conv-GRU. It is constituted by the reset gate, update gate, and main body of Conv-GRU.

The calculation formula is shown in Equation (1):

$$
\begin{aligned}
z_t &= \sigma(W_{xz} * x_t + W_{hz} * h_{t-1}) \\
r_t &= \sigma(W_{xr} * x_t + W_{hr} * h_{t-1}) \\
h'_t &= tanh(W_x * x_t + r_t \circ (W_h * h_{t-1})) \\
h_t &= (1 - z_t) \circ h'_t + z_t \circ h_{t-1}
\end{aligned}
\tag{1}
$$

where $\sigma$ is mean Sigmoid activation function, tanh is tanh activation function, $\circ$ is element-wise multiplication, $*$ is convolution, $x_t$ is input features extracted by backbone, and $W_{xz}$, $W_{hz}$, $W_{xr}$, $W_{hr}$, $W_x$, $W_h$ are the 2D convolutional kernels whose parameters are optimized end-to-end.

Temporal Attention Gated Recurrent Unit (TA-GRU). Different from the original Conv-GRU, we modified it to make it extend to drone video object detections. The overall Temporal Attention Gated Recurrent Unit (TA-GRU) architecture is shown in Figure 1. We used it to propagate temporal features to more effectively retain temporal information; we chose deformable transformer layer to replace the original convolutional layer and added the temporary processing module (temporal alignment and fusion) to aggregate input and temporal features. The deformable transformer layer can enable the model

to focus more effectively on target areas, and it is better at handling temporal inputs than traditional convolutional layers, resulting in improved performance compared to traditional convolutional layers. In TA-GRU module, the temporal features are propagated frame-by-frame between inputs to improve each frame appearance features. The final outputs will be batch inputs. The specific formula is shown in Equation (2):

$$
\begin{aligned}
Z_t &= \sigma(cross\_attn(W_{hz}, self\_attn(W_{xz}, x_t), H_{t-1})) \\
R_t &= \sigma(cross\_attn(W_{hr}, self\_attn(W_{xr}, x_t), H_{t-1})) \\
H'_t &= tanh((self\_attn(W_x, x_t) + Tem\_prc(R_t \circ H_{t-1})) \\
H_t &= (1 - Z_t) \circ H'_t + Z_t \circ H'_t
\end{aligned}
\tag{2}
$$

where $\sigma$ is mean Sigmoid activation function, tanh is tanh activation function, $\circ$ is element-wise multiplication, $self\_attn$ is self-deformable attention, $cross\_attn$ is cross-deformable attention, $Tem\_prc$ is the mean after temporal processing on temporal features, $x_t$ is input features extracted by backbone, and $W_{xz}$, $W_{hz}$, $W_{xr}$, $W_{hr}$, $W_x$ are the weight matrix of deformable attention whose parameters are optimized end-to-end.

Temporal Deformable Transformer Layer (TDTL). To our knowledge, there are a lot of tiny objects in drone videos, which will introduce much background. Previous work [17] has addressed this issue by adding a transformer layer at the neck of the model to enhance the features extracted from the backbone. However, the general transformer layer [18] will introduce much computation overhead. The viewpoint in DETR [19] suggests that the more relevant area to the target area is often its nearby area. Furthermore, a video object detector was built using a deformable transformer within TransVOD [3] and attained satisfactory detection outcomes. Therefore, we utilized a deformable transformer layer to build our Temporal Deformable Transformer Layer (TDTL). This module will make the model pay more attention on target areas to improve the features. As shown in Figure 3, the deformable transformer layer only assigns a small, fixed number of keys for each query. Given an input feature map $x \in R^{C \times H \times W}$, let $i$ index a 2D reference point $p_i$. The deformable attention feature is calculated by Equation (3):

$$
DeformAttn(x, p_i) = \sum_{n=1}^{N} W_n \left[ \sum_{k=1}^{K} A_{nik} W'_n x(p_i + \Delta p_{nik}) \right]
\tag{3}
$$

where $n$ indexes the attention head, $k$ indexes the sampled keys, $\Delta p_{nik}$ and $A_{nik}$ denote the sampling offset and attention weight of the $k$th sampling point in the $n$th attention head, respectively, and the scalar attention weight $A_{nik}$ lies in range [0, 1], normalized by $\sum_{k=1}^{K} A_{nik} = 1$.

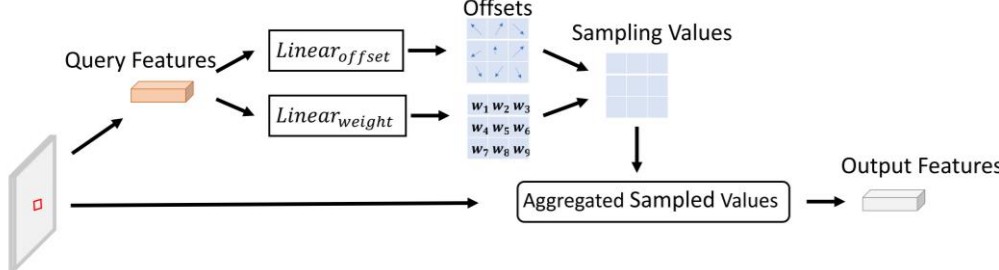

**Figure 3.** Architecture of Temporal Deformable Transformer Layer (TDTL). To reduce huge computation overhead on a typical transformer layer, the deformable transformer layer only attends to a small set of key sampling points around the reference.

We chose self-deformable attention to improve the attention on target areas of the input features, then used cross-deformable attention to complete the interaction with temporal features. By implementing this approach, our model becomes more adept at emphasizing the features of the current frame during the update of temporal features while also giving due consideration to the previous temporal features when determining which information

should be preserved. This enhanced flexibility enables our network to focus more precisely on the specific areas of interest.

DeformAlign. We noticed that same object features are usually not spatially aligned across frames due to video motion. Without proper feature alignment before aggregation, the object detector may generate numerous false recognitions and imprecise localizations. Therefore, recent works [1,8] have utilized deformable convolution [20] to compute offset caused by movement between different frames to align different frame features. The architecture of the DeformAlign module is shown in Figure 4. Different from the deformable convolution module, we needed model motion in different frames so we used an extra convolution layer to simply fuse different frame features. Then, we used two different convolutions to compute the offsets and corresponding weights by choosing the fused features as inputs and utilized the offsets and weights to align neighboring frame features to the reference frame features. Given the prevalence of small targets in drone imagery, where the inter-frame motion of these targets may not be substantial, we found that a single layer of deformable convolution was sufficient to effectively capture their motion information.

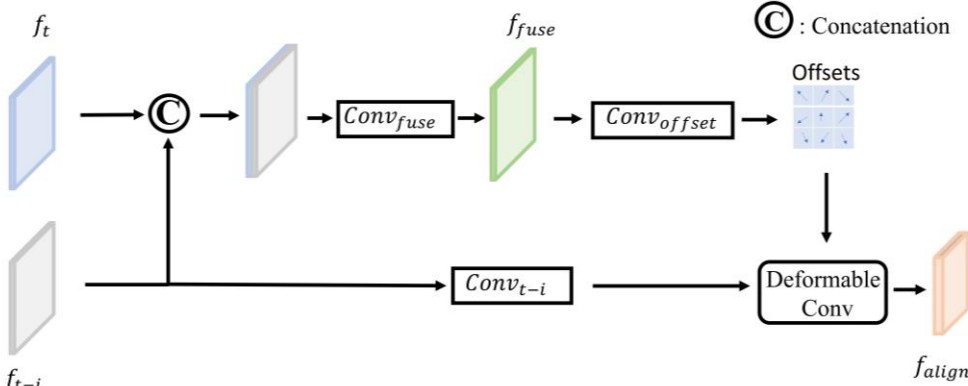

**Figure 4.** Architecture of DeformAlign. We used an extra convolution layer to simply fuse different features connected by channel. We used bilinear interpolation in a deformable convolution module to align the features of neighboring frames to the reference frame.

Two-dimensional convolution samples were positioned on a uniformly spaced grid $R$, and we used weight $W$ to sum the sampling values. For example, when we use a convolution that $kernel\_size = 3 \times 3, stride = 1$ to compute the pixel at position $p_0$, we can obtain the corresponding new value on feature map $y$ by following Equation (4):

$$y(p_0) = \sum_{i=1}^{N} W_{p_i} \cdot x(p_0 + \Delta p_i) \tag{4}$$

where $N = kernel - size$, $\Delta p_i = \{(-1,-1),(-1,0) \cdots (1,1)\}$, $W_{p_i}$ is the corresponding weight at $p_0 + \Delta p_i$.

Deformable convolution introduces two additional convolutional layers to adaptively calculate offset $\Delta p_n$ and weight $\Delta w_n$. We can compute the aligned pixel at $p_0$ by following Equation (5):

$$y_{align}(p_0) = \sum_{i=1}^{N} W_{p_i} \cdot x(p_0 + \Delta p_i + \Delta p_n) \cdot \Delta w_n \tag{5}$$

It uses bilinear interpolation to achieve the process of $p_0 + \Delta p_i + \Delta p_n$.

Temporal Attention and Temporal Fusion Module (TA-Fusion). TCE-Net [1] notices that there are different contributions to reference frame features in different frame features. The goal of temporal attention is to compute frame similarity in an embedding space to focus on 'when' it is important given neighboring frames. Intuitively, at location $p$, if the aligned features $f_{align}$ are close to reference features $f_t$, they should be assigned higher weights. Here, dot product similarity metric is used to measure the similarity. Additionally, temporal fusion is proposed to aggregate features from neighboring frames to model temporal context. We used a $1 \times 1 \times C$ convolutional network to fuse the aligned temporal

features with the features of the current frame. During the training process, the parameters of the fusion network were adaptively updated, enhancing the efficiency of feature fusion in our model and improving overall performance.

The weights of temporal attention map are estimated by Equation (6):

$$M_t(p) = \sigma(f_{align}(p) \cdot f_t(p)) \tag{6}$$

where $\sigma$ is Sigmoid activation function. The architecture of the temporal attention and temporal fusion module is shown in Figure 5.

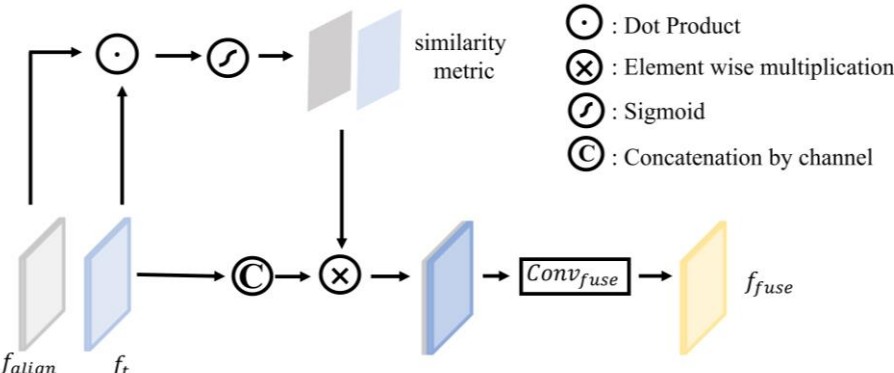

**Figure 5.** Architecture of Temporal Attention and Temporal Fusion module. We used dot product to measure the similarity of $f_{align}$ and $f_t$. Then, we used the similarity metric to assign different weights for each frame. Finally, we chose a $1 \times 1 \times C$ convolution layer to aggregate features.

As shown in Figure 5, the temporal attention maps have the same spatial size with $f_t$ and are then multiplied in a pixel-wise manner to the original aligned features $f_{align}$.

## 4. Experiments

### 4.1. Training Dataset and DETAILS

Training Dataset. We trained and tested on the VisDrone2019-VID dataset [21], which includes 288 video clips taken by the UAV platform at different angles and heights. All videos are fully annotated with object bounding box, object category, and tracking IDs. There are 10 object categories ('pedestrian', 'people', 'bicycle', 'car', 'van', 'truck', 'tricycle', 'awning-tricycle', 'bus', 'motor') consisting of 261,908 images, 24,201 for training images, 2846 for validate images, and 6635 for test images. Unlike other general video object detection datasets, there are a lot of tiny objects and severe appearance deterioration in it. Thus, we needed a video object detection method that could aggregate extensive tiny object features to solve the appearance deterioration. Mean average precision (mAP) (average of all 10 IoU thresholds, ranging from [0.5:0.95]) and AP50 were used as the evaluation metric.

Implementation Details. Our modules rely on one NVIDIA RTX3090 GPU for both training and testing. Additionally, our experiments show that the diversity of the video clips in VisDrone2019-VID is significantly lower when compared to ImageNet-VID. Hence, it was necessary for us to perform additional data processing on VisDrone2019-VID. Referring to the method in TCE-Net [1], we chose a temporal stride predictor that took the differences between features $t$ and features $k$ to select which frames to aggregate. This predictor takes the differences between features $t$ and features $k$, i.e., $(f_t - f_k)$, as input and predicts the deviation score between frame $t$ and frame $k$. The deviation score is formally defined as the motion intersection-over-union (IoU). If IoU < 0.5, the temporal stride is set to 1. If 0.5 < IoU < 0.7, the temporal stride is set to 2. Furthermore, if IoU > 0.7, the temporal stride is set to 4. Inspired by FGFA [4], we firstly used VisDrone2019-DET to pretrain our model by setting $batch\_size = 1$. We then used the pretrained model weights as the resume model to continue training on VisDrone2019-VID. Because the VisDrone2019-VID training set is a bit small, we only trained the model on VisDrone2019-VID trainset for 70 epochs, and

the first 2 epochs were used for warm-up. We used an SGD optimizer for training and $5 \times 10^{-4}$ as the initial learning rate with the cosine learning rate schedule. The learning rate of the last epoch decays to 0.01 of the initial learning rates. Considering the small objects in the drone image, we assigned the size of the image to 1280 pixels. The important parameters of the training process were set, as shown in Table 1.

**Table 1.** Training parameter setting table.

| Parameters | Setup |
| --- | --- |
| Epochs | 70 |
| Batch Size | 4 |
| Image Size | $1280 \times 1280$ |
| Initial Learning Rate | $2 \times 10^{-4}$ |
| Final Learning Rate | $2 \times 10^{-6}$ |
| Momentum | 0.937 |
| Weight-Decay | $5 \times 10^{-4}$ |
| Image Scale | 0.6 |
| Image Flip Left-Right | 0.5 |
| Mosaic | 0 |
| Image Translation | 0.2 |
| Image Rotation | 0.2 |
| Image Perspective | $2 \times 10^{-5}$ |

Data Analysis. Based on our past experience, it is crucial to analyze the dataset thoroughly before designing and training a model in order to construct an effective one. Upon reviewing the VisDrone2019-VID dataset, we observed the presence of numerous small objects, as well as some appearance deterioration such as part occlusion, motion blur, and video defocus. Therefore, there is an urgent need to develop a simple yet effective VOD framework that can be fully end-to-end.

In Figure 6, there are numerous objects smaller than 4 pixels. While these objects aided in training our temporal aggregated module, they should not be included in the computation of the model loss function. Typical methods for handling the ignore regions in the VisDron2019-VID dataset involve replacing them with gray squares. However, our experiments show that this approach can result in a loss of image information, particularly in UAV images, which is not conducive to training the temporal aggregated module. To prevent our model from detecting ignore regions and to retain useful training information, we chose to map the predicted bounding box back to the original images and set the intersection-over-union (IoU) to 0.7 about ground truth bounding box and ignore regions, thus excluding the ignore regions from loss calculation. This method has proven to be more effective than simply replacing the ignored regions with gray squares, resulting in a 0.72% increase in mean average precision (mAP).

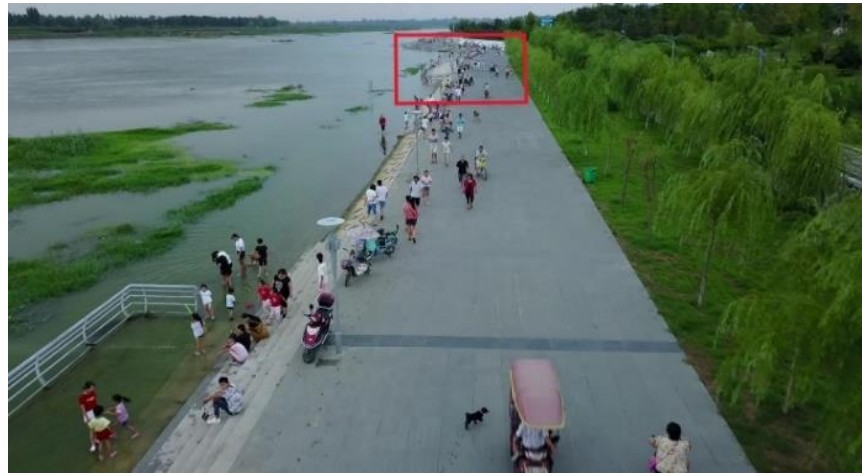

**Figure 6.** Illustration of the 'ignore region' in the VisDrone2019-VID dataset.

### 4.2. Comparisons to State-of-the-Art

Table 2 shows the comparison of TA-GRU YOLOv7 with other state-of-the-art methods. With the use of temporal post-processing techniques, D&T adopts a well-designed tubelet rescore technique, while others use Seq-NMS. The results demonstrate that our TA-GRU module is effective when compared to image-based object detectors such as YOLOv7.

**Table 2.** Object detection results on VisDrone2019-VID.

| Methods | mAP (%) | AP50 (%) | Aggregate Frames | FPS |
|---|---|---|---|---|
| TA-GRU YOLOv7 | 24.57 | 48.79 | 2 | 24 |
| YOLOv7 [22] | 18.71 | 40.26 | - | 45 |
| TA-GRU YOLOX | 19.41 | 40.59 | 2 | - |
| YOLOX [23] | 16.86 | 35.62 | - | - |
| TA-GRU YOLOv7-tiny | 0.165 | 0.296 | 2 | - |
| YOLOv7-tiny | 0.103 | 0.212 | - | - |
| FGFA [4] | 18.33 | 39.71 | 21 | 4 |
| STSN [8] | 18.52 | 39.87 | 27 | - |
| D&T [24] | 17.04 | 35.37 | - | - |
| FPN [25] | 16.72 | 39.12 | - | - |
| CornerNet [26] | 16.49 | 35.79 | - | - |
| CenterNet [27] | 15.75 | 34.53 | - | - |
| CFE-SSDv2 [28] | 21.57 | 44.75 | - | 21 |

Table 2 shows that TA-GRU YOLOv7 achieves a higher mean average precision (mAP) than YOLOv7, with an improvement of 5.86% mAP. Moreover, the computational overhead introduced by our method is small, which provides strong evidence for its effectiveness. Compared with FGFA (18.33% mAP), TA-GRU YOLOv7 obtains 24.57% mAP, outperforming it by 6.24%. Furthermore, TA-GRU YOLOv7 only aggregates a temporal feature and reference frame feature, while FGFA is 21. Additionally, with a deformable convolution detector and temporal post-processing, STSN obtains 18.52% mAP. However, TA-GRU YOLOv7 obtains 24.57% mAP, which is about 6.05% higher than it. The detection effect of some scenes is shown in Figure 7.

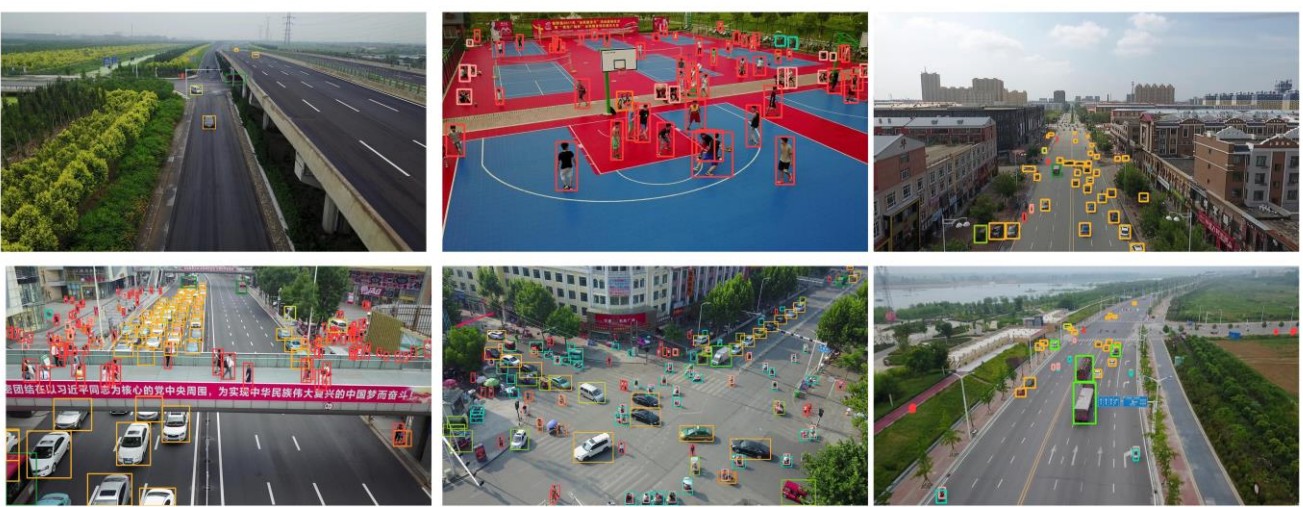

**Figure 7.** Examples of the detection effect.

Table 3, presented below, illustrates the detection outcomes of our model across various categories in the VisDrone2019-VID dataset. Our model has achieved outstanding detection performance across the vast majority of categories.

**Table 3.** The detection of our network on various categories on VisDrone2019-VID dataset.

| Classification | mAP (%) | AP50 (%) | P | R |
|---|---|---|---|---|
| all | 24.57 | 48.79 | 0.577 | 0.513 |
| pedestrian | 29.1 | 68.1 | 0.658 | 0.668 |
| people | 18.5 | 49.7 | 0.56 | 0.579 |
| bicycle | 30.1 | 65.0 | 0.572 | 0.663 |
| car | 40.1 | 63.6 | 0.737 | 0.636 |
| van | 26.0 | 43.7 | 0.725 | 0.422 |
| truck | 32.1 | 56.1 | 0.636 | 0.578 |
| tricycle | 16.7 | 37.8 | 0.537 | 0.428 |
| awning-tricycle | 12.9 | 25.8 | 0.496 | 0.261 |
| bus | 25.4 | 33.4 | 0.368 | 0.345 |
| motor | 14.3 | 42.3 | 0.481 | 0.547 |

*4.3. Ablation Study and Analysis*

In order to evaluate the effectiveness of our proposed methods, we conducted a series of experiments to analyze the impact of key components. This section provides a detailed analysis of our findings, including experimental results and insights into how each component contributes to the overall success of our approach.

Ablation for TA-GRU YOLOv7

Table 4 compares our TA-GRU YOLOv7 with the single-frame baseline and its variants.

**Table 4.** Accuracy and runtime of different methods on VisDrone2019-VID validation. The runtime contains data processing, which is measured on one NVIDIA RTX3090 GPU.

| Methods | (a) | (b) | (c) | (d) | (e) | (f) |
|---|---|---|---|---|---|---|
| Conv-GRU? | | √ | √ | √ | | |
| TA-GRU? | | | | | √ | √ |
| DeformAlign? | | | √ | √ | √ | √ |
| Temporal Attention and Temporal Fusion module? | | | | √ | √ | √ |
| end-to-end training? | √ | √ | √ | √ | √ | |
| mAP (%) | 18.71 | 16.55 | 20.03 | 23.96 | 24.57 | 23.82 |
| AP50 (%) | 40.26 | 37.29 | 41.68 | 47.61 | 48.79 | 47.23 |

Method (a) is the single-frame baseline. It has a mAP of 18.71% using YOLOv7. It outperforms the video detector, FGFA, by 0.38%. This indicates that our baseline is competitive and serves as a valid reference for evaluation.

Method (b) is a naive feature aggregation approach and a degenerated variant of TA-GRU YOLOv7, which uses Conv-GRU to aggregate temporal features. The variant is also trained end-to-end in the same way as TA-GRU YOLOv7. The mAP decreases to 16.55%, 2.16% shy of baseline (a). This indicates that using traditional feature fusion networks to directly aggregate complex drone video features can potentially introduce background interference.

Method (c) adds the DeformAlign module into (b) to align neighboring frame features to the reference frame features. It obtains a mAP of 20.03%, 1.32% higher than that of (a) and 3.48% higher than that of (b). This result suggests that when features are aligned to the same spatial position, it enhances the fusion of effective features in the fusion network. However, introducing noise remains inevitable.

Method (d) adds the temporal attention and temporal fusion module to (c). It increases the mAP score from 20.03% to 23.96%. Figure 8 shows that images with distinct appearance features are assigned varying weights depending on how similar they are to the features of the reference frame. This also effectively eliminates the impact of noise information from adjacent frames on the features of the reference frame.

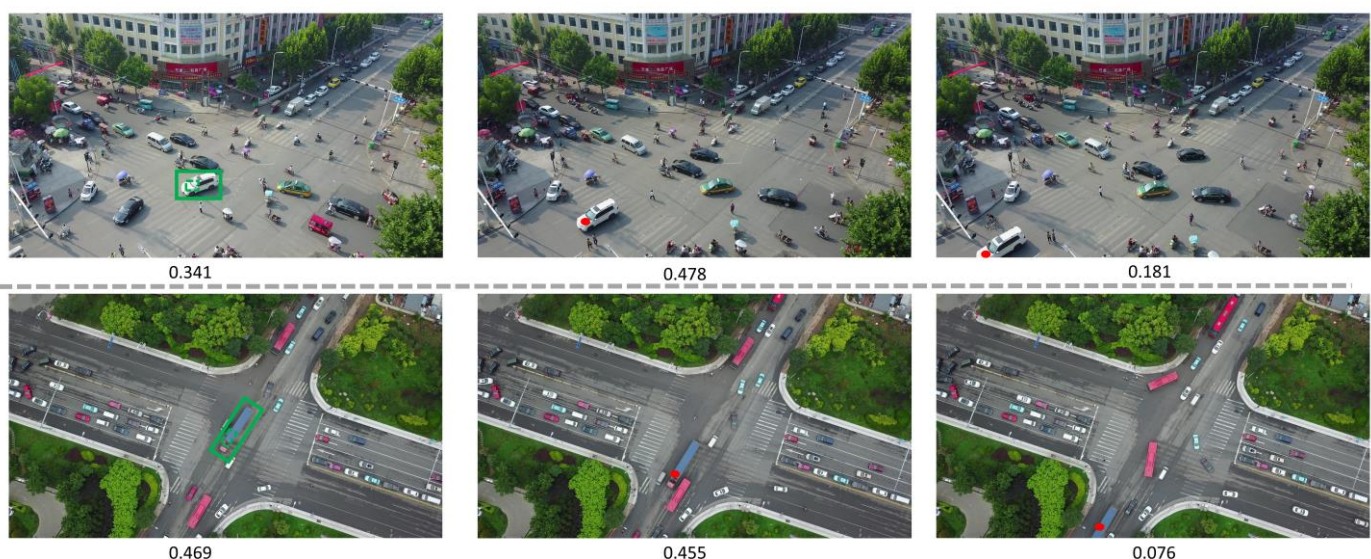

**Figure 8.** Images with distinct appearance features are assigned varying weights. The weight is determined by both the distance and similarity to the reference frame.

Method (e) is the proposed TA-GRU YOLOv7 method, which uses deformable attention to replace the convolution layer in (d). It increases the mAP score from 23.96% to 24.57%. It suggests that the deformable attention make model pays more attention to target areas to effectively promote the information from nearby frames in feature aggregation. The proposed TA-GRU YOLOv7 method improves the overall mAP score by 5.86% compared to the single-frame baseline (a).

Method (f) is a degenerated version of (e) without using end-to-end training. It takes the feature and the detection sub-networks from the single-frame baseline (a). During training, these modules are fixed and only the embedding temporal extracted module is learnt. It is clearly worse than (e). This indicates the importance of end-to-end training in TA-GRU YOLOv7.

## 5. Conclusions

This work presents an accurate, simple yet effective VOD framework in a fully end-to-end manner. Because our approach focuses on improving feature quality, it would be complementary to existing single frameworks for better accuracy in video frames. Our primary contribution is the integration of recurrent neural networks, transformer layers, and feature alignment and fusion modules. Ablation experiments show the effectiveness of our modules. Together, the proposed model not only achieves a 24.57% mAP score on VisDorne2019-VID, but also reaches a running speed of 24 frames per second (fps). However, more annotation data and precise motion estimation may be beneficial for improvements. Indeed, our module currently lacks proficiency in handling long-term motion information, and the degradation of appearance characteristics in various objects within UAV images presents a challenge for our module's ability to effectively learn temporal information. Addressing this issue is a key objective for our next stage of development.

**Author Contributions:** Conceptualization, Z.Z.; methodology, Z.Z.; software, Z.Z.; validation, Z.Z.; formal analysis, Z.Z. and X.Y.; investigation, Z.Z. and X.Y.; resources, X.Y.; data curation, Z.Z. and X.Y.; writing—original draft preparation, Z.Z.; writing—review and editing, Z.Z. and X.Y.; visualization, Z.Z.; supervision, X.C.; project administration, X.Y.; funding acquisition, X.Y. All authors have read and agreed to the published version of the manuscript.

**Funding:** This work was supported by the National Natural Science Foundation of China under Grant 61973309 and the Natural Science Foundation of Hunan Province under Grant 2021JJ20054.

**Data Availability Statement:** The data presented in this study are openly available in The Vision Meets Drone Object Detection in Video Challenge Results (VisDrone-VID2019) at https://github.com/VisDrone/VisDrone-Dataset, accessed on 29 May 2023.

**Conflicts of Interest:** The authors declare no conflict of interest. The funders had no role in the design of the study; in the collection, analyses, or interpretation of data; in the writing of the manuscript, or in the decision to publish the results.

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
