# Peer review of "Object Detection in Drone Video with Temporal Attention Gated Recurrent Unit Based on Transformer"

_drones, doi:10.3390/drones7070466_

Round 1
Reviewer 1 Report
Comments and Suggestions for Authors
There are a few instances in the manuscript where the name of the used public dataset, VisDrone2019-VID, is misspelled. For instance, it is written as "VisrDron2019-VID" (line 298) and "VisDroen2019-VID" (line 332). Additionally, the citation of the dataset could be improved to ensure accuracy and proper referencing.
The word module is misspelled in line 172 (moule).
Reviewer 2 Report
Comments and Suggestions for Authors
Dataset Limitation: The work only evaluates the proposed method on the VisDrone2019-VID dataset, which may not fully represent all possible scenarios and challenges in UAV-based object detection. The generalizability of the proposed method to other datasets and real-world scenarios is not extensively demonstrated.
Lack of Comparative Analysis: While the work claims to achieve state-of-the-art performance on the VisDrone2019-VID dataset, it does not provide a comprehensive comparison with other existing methods or baselines. A comparative analysis would have helped in better understanding the relative strengths and weaknesses of the proposed approach.
Evaluation Metrics: The work mentions an improvement in mean Average Precision (mAP) by 5.86%, but it does not provide detailed information about other evaluation metrics or how the proposed method performs in terms of different object classes or challenging scenarios. A more thorough analysis of various evaluation metrics would have provided a better assessment of the proposed method's performance.
Computational Cost: Although the work claims that the proposed module has a negligible extra computational cost, it does not provide detailed information or quantitative analysis regarding the computational requirements of the proposed method compared to the baseline. The actual impact on computational resources, such as processing power and memory, is not explicitly addressed.
The current state of the art in UAV-based object detection is not rich in terms of comprehensive advancements and robust solutions. While there have been notable efforts to address the challenges posed by appearance deterioration in images captured by drones, the existing literature lacks a multitude of innovative and highly effective approaches. you can add more some references.
Generalization to Other Object Detectors: The work mentions that the proposed module can easily be used to enhance existing static image object detectors for video object detection. However, it does not provide sufficient information or empirical evidence to support this claim. The effectiveness of the proposed module with different object detectors or architectures is not thoroughly investigated.
Lack of Insight into Feature Alignment & Fusion Modules: The work highlights the integration of recurrent neural networks, transformer layers, and feature alignment & fusion modules as a major contribution. However, it does not provide detailed explanations or insights into the design and functioning of these modules. A more in-depth analysis of these modules would have helped in better understanding their effectiveness and potential limitations.
Overemphasis on Performance Metrics: While the work primarily focuses on achieving high detection accuracy and real-time performance, it does not address other important aspects such as robustness to occlusion, scale variation, or adverse weather conditions, which are commonly encountered in UAV-based object detection. The limitations of the proposed method in handling these challenges are not discussed.
External Factors: The work briefly mentions that more annotation data and precise motion estimation may benefit improvements. However, it does not delve into the limitations or challenges associated with obtaining accurate annotations or motion estimation in drone videos. The impact of these external factors on the proposed method's performance is not thoroughly addressed.
Comments on the Quality of English Language
need to be improved
Round 2
Reviewer 2 Report
Comments and Suggestions for Authors
I wanted in first revision to make the code online, however it seems the authors did not imply that, so I want to see the code, becuase I have some doubts about the results.
Comments on the Quality of English Language
need minor editing
Round 3
Reviewer 2 Report
Comments and Suggestions for Authors
the code need to be send in Zip file because the link sent by authors not working.
Comments on the Quality of English Languageneed to be improved.
Author Response
Thank you for your reply. Due to our inability to register for a Google Drive account, we have sent the zip file of our code to the editorial assistant, who will help us forward it to you via email.